# Risk of Antimicrobial Resistant Non-Typhoidal *Salmonella* during Asymptomatic Infection Passage between Pet Dogs and Their Human Caregivers in Khon Kaen, Thailand

**DOI:** 10.3390/antibiotics9080477

**Published:** 2020-08-04

**Authors:** Xin Wu, Sunpetch Angkititrakul, Allen L. Richards, Chaiwat Pulsrikarn, Seri Khaengair, Amphone Keosengthong, Supatcharee Siriwong, Fanan Suksawat

**Affiliations:** 1Ecotoxicology, Natural Resources and Environment Project, Khon Kaen University, Khon Kaen 40002, Thailand; wuxinddl@hotmail.com (X.W.); sunpetch@kku.ac.th (S.A.); serkan@kku.ac.th (S.K.); 2College of Veterinary Medicine, Yunnan Agricultural University, Kunming 650051, China; 3Department of Preventive Medicine and Biostatistics, Uniformed Services University of the Health Sciences, Bethesda, MD 20814, USA; Allen.Richards@comcast.net; 4National Institute of Health, Department of Medical Sciences, Ministry of Public Health, Nonthaburi 11000, Thailand; chaiwat.p@dmsc.mail.go.th; 5Department of Veterinary Medicine, Faculty of Agriculture, National University of Laos, Vientiane 7322, Lao People’s Democratic Republic; Keosengthong1990@gmail.com; 6Synchrotron Light Research Institute (Public Organization), Nakhon Ratchasima 30000, Thailand; supatcharee@slri.or.th

**Keywords:** antimicrobial resistance, non-typhoidal *Salmonella*, asymptomatic infection passage, pet dogs, humans

## Abstract

To explore the risk of antimicrobial resistant (AMR) non-typhoidal *Salmonella* during asymptomatic infection passage between pet dogs and human caregivers in Khon Kaen, Thailand, one hundred forty paired fecal samples (*n* = 280) were obtained from companion dogs and their human caregivers, interviewed from 140 households during 2019–2020. The purified *Salmonella* isolates were serotype-identified and tested for antimicrobial resistance against ampicillin, ciprofloxacin, chloramphenicol, nalidixic acid, streptomycin, sulfamethoxazole–trimethoprim, and tetracycline. The homologous *Salmonella* isolate pairs (suggesting *Salmonella* infections may have been due to passage between each one of the pair, or derived from the same source) were subsequently characterized by serotype screening, pulsed field gel electrophoresis (PFGE), and Synchrotron Fourier transform infrared spectroscopy (SR-FTIR). The *Salmonella* prevalence observed in dogs, 12.86% (18/140), was not significantly different from that observed in humans, 17.86% (25/140) using McNemar’s test. The AMR patterns (the patterns among the isolates of pet dogs and caregivers) and the serotypes (thirteen serotypes with 18 isolates from pet dogs plus thirteen serotypes with 25 isolates from humans) between pet dogs and humans were not significantly different using Pearson’s chi-squared test. The homologous *Salmonella* isolates from the *Salmonella*-present households was 5.13% (2/39). This study demonstrated that the hypothesis that there is a high risk of *Salmonella* infection passage between dogs and humans with close contact in Khon Kaen is doubtful. Only 5.13% of homologous *Salmonella* isolates from *Salmonella*-present households were found in Khon Kaen, Thailand, although the prevalence of *Salmonella*-positive samples, serotypes, and antimicrobial resistance patterns were quite similar among the two populations.

## 1. Introduction

Antimicrobial resistance (AMR) has been a global problem for managing the health care of people and animals. AMR pathogens make the problem worse, because these pathogens can transmit AMR genes and cause infections in humans and animals [1]. The measures of transmissibility of AMR pathogens between animals and humans could provide information to discern an important route of how AMR pathogens get to humans. However, it is hard to ascertain the transmission direction (whether it is transmitted from animals to humans or from humans to animals). As such, an association investigation would be a better choice.

*Salmonella* species (non-typhoidal) are a leading cause of 550 million diarrheal diseases worldwide each year [2]. Additionally, AMR *Salmonella* infections makes the problem worse, as they can lead to treatment ineffectiveness, increasing the risk of blood infections [3], and the subsequent spread of AMR [4]. A wide variety of *Salmonella* (non-typhoidal) agents were isolated and reported from the food chain [5]. Unfortunately, there is a lack of credible data for pets as a source of salmonellosis, despite salmonellosis from pets being suggested as an important AMR transmission route [6]. Pet dogs may be infected with *Salmonella* from eating contaminated food and contact with infected feces or other infected animals. Subsequently, humans may become infected after coming into contact with a sick dog. One survey reported that of the 128 *Salmonella* infected patients, 107 (84%) ill people had contacted with dogs prior to getting sick [7]. Several cross-sectional studies also reported that the risk of salmonellosis in humans is increased if their pet dogs had salmonellosis without overt clinical signs due to their close relationship [8,9]. However, despite data suggesting a significant *Salmonella* transmission route between pet dogs and humans, reliable quantitative data on AMR *Salmonella* transmission via the pet dog pathway are still lacking [10].

According to the World Health Organization (WHO) database [11], the Southeast Asia region has a high burden of non-typhoidal salmonellosis. The report indicates that the Southeast Asia region contributes 20.69% cases (16.28 million/78.7 million) and 26.93% deaths (15,929/59,153) of the global non-typhoidal salmonellosis. Similarly, another research report estimated worldwide cases and deaths caused by non-typhoidal *Salmonella* at 93.8 million cases, with 155,000 deaths occurring each year, and that in Southeast Asia, 22.8 million cases and 37,600 death occur [5]. On the other hand, it has been reported that 11.6% of diarrheal and 13.2% of non-diarrheal dogs were infected by *Salmonella* in Northeastern Thailand [9]. The estimated dog population in Thailand is about 8.5 million [12]. Thus, there appears to be a high risk of *Salmonella* infection among pet dog sources. However, there has been no research reported on the association of AMR *Salmonella* between pet dogs and human care givers in Khon Kaen, in Northeast Thailand.

In order to investigate the association between *Salmonella* infections (and more importantly AMR *Salmonella* infections) between pet dogs and humans in Khon Kaen, Thailand, matched pair fecal samples of pet dogs and their caregivers were assessed. The paired fecal samples from pet dogs and their caregivers were investigated in Khon Kaen for the presence, serotype identity, and antimicrobial susceptibility of *Salmonella*. In addition, the quantitative presence of homologous *Salmonella* isolates among paired individuals (pet dogs and their human caregivers) was ascertained.

## 2. Material and Methods

### 2.1. Sample Size Calculation

The formula: $n={\left(z\right)}^{2}p\left(1-p\right)/d^{2}$ was applied to calculate the required sample size. In the formula, ‘*n*’ represents the required sample size; ‘*z*’ represents the level of confidence according to a standard normal distribution (for a level of confidence of 95%, *z* = 1.96); ‘*p*’ represents the expected prevalence, and ‘*d*’ represents allowable error (here, it was set at 5%).

Approximately 5% of people in Thailand are infected with *Salmonella*, as reported by Sirinavin [13]. The prevalence of *Salmonella* infected pet dogs is approximately 10%, as reported by Polpakdee [9]. The required sample sizes for humans and pet dogs were determined to be 73 and 138, respectively, according to the expected prevalence and allowable error. For paired dogs and humans, there was no preliminary study for *Salmonella* prevalence between the two populations. For this study, the required paired sample size was set according to the required pet dog sample size because it was larger than the human sample size required.

### 2.2. Sample Collection

One hundred and forty households were randomly interviewed from five villages (31 households from Bua Simma village, 24 from Mo Sum Soong village, 29 from Non Udom village, 44 from Noun Kung Soen village, and 12 from Nai Muang village) in Khon Kaen Province, Thailand, from 2019 to 2020 (random sampling). For the 140 household interviews, one pet dog and its closest caregiver were requested for sampling in pairs (*n* = 280). Contact means physical contact, and closest means the human caregiver who most often fed and cleaned the pet within the household. All subjects (both pet dogs and humans) were investigated for their history of antimicrobial usage for the past year. However, the history of antimicrobial usage was often incomplete, because many human volunteers could not provide information on the types of antimicrobials used. All samples were collected from asymptomatic individuals (both pet dogs and humans). The carriage of *Salmonella* among the individuals evaluated from the five villages were sporadic without apparent outbreak associations (i.e., no enteritis in groups occurred within the five villages, as indicated by the village health administrator). Fecal samples were collected using sterile rectal swabs for pet dogs and 10 g of feces using sterile plastic containers for human caregivers. All samples were matched and transported in an icebox. *Salmonella* isolation was subsequently performed at the laboratory. The study was guided and approved by the Institutional Animal Care and Use Committee of Khon Kaen University, No. IACUC-KKU-98/61, and Khon Kaen University Ethics Committee for Human Research, No. HE602286.

### 2.3. Salmonella Isolation

All fecal samples were immediately processed after transport for *Salmonella* isolation following the International Organization for Standardization (ISO) standard 6579-1:2002 at the Khon Kaen University Veterinary Teaching Hospital laboratory [14]. All swab samples were pre-enriched in 9 mL buffered peptone water (BPW) (HiMedia Laboratories, Mumbai, Maharashtra, India) at 37 °C for 24 h, and the fecal samples were transferred by swabs into 9 mL BPW. Then, each aliquot of 0.1 mL was selectively enriched on Modified Semisolid Rappaport Vassiliadis Medium plates (Becton Dickinson, MSRV, Franklin Lake, NJ, USA) at 42 °C for 24 h. The turbid-grey material from MSRV was streaked onto xylose lysine deoxycholate agar (XLD) plates (Becton Dickinson) at 37 °C for 24 h, and secondly selected on Hektoen Enteric agar (HE) plates (Becton Dickinson) at 37 °C for 24 h. The presumptive colonies were identified as *Salmonella* by their black color.

### 2.4. Salmonella Serotype Identification

Three presumptive colonies from selective media with each sample were selected and assessed to determine their serotype(s). If the three colonies were of the same serotype, they would be regarded as one isolate. If different serotypes from the three presumptive colonies from an individual sample were identified, then all unique colonies were assessed and recorded. All presumptive colonies were biochemically identified by Triple Sugar Iron agar (TSI) (Becton Dickinson) and Motility Indole Lysine (MIL) medium (Becton Dickinson). The O (somatic) and H (flagellar) antigens of all isolates were characterized by agglutination with hyperimmune sera (S&A Reagents Lab Ltd., Bangkok, Thailand), at the WHO National Salmonella and Shigella Center for serotype identification. The serotypes were assigned according to the Kauffmann–White scheme [15].

### 2.5. Antimicrobial Susceptibility Testing

The *Salmonella* isolates were analyzed for antimicrobial susceptibility to ampicillin (AMP) 10 μg, ciprofloxacin (CIP) 5 μg, chloramphenicol (CHL) 30 μg, nalidixic acid (NA) 30 μg, streptomycin (STR) 10 μg, sulfamethoxazole–trimethoprim (SXT) 25 μg and tetracycline (TET) 30 μg by the disk diffusion method (BD Diagnostics, Sparks, MD, USA), following the Clinical and Laboratory Standards Institute guidelines [16].

### 2.6. The Statistical Analysis of Salmonella Asymptomatic Infection Passage between Paired Dogs and Their Caregivers

To assay the epidemiological relatedness of *Salmonella* isolates from asymptomatic infection passage between the two populations, the presence of *Salmonella*-positive samples was statistically analyzed by McNemar’s test. The serotypes and antimicrobial patterns of *Salmonella* isolates were statistically analyzed by Pearson’s chi-squared test. The statistical analysis was performed using statistical software SPSS version 17.0 (SPSS Inc, Chicago, IL, USA), and *p* < 0.05 was considered significant.

### 2.7. The Determination of Homologous Salmonella Isolates

To determine the homologous *Salmonella* isolate pairs, serotypes of all pet dog and human source isolates were screened for serotype-matched pairs of isolates (the isolates from a pet dog and its human caregiver belonging to the same serotype). Then, the serotype-matched pairs of isolates were tested by pulsed field gel electrophoresis (PFGE) and further confirmed by Synchrotron Fourier transform infrared spectroscopy (SR-FTIR) for homologous *Salmonella* isolate pairs. FTIR is a rapid and accurate detection technique, providing reliable data for bacterial strain level epidemiological research [17,18]. The detailed assessment of homologous *Salmonella* isolate pairs allowed for the demonstration of whether or not the AMR *Salmonella* spp. infected dogs and the infected humans of the same household were related (whether or not the infections arise from the passage between each other or from the same source).

The PFGE was performed following the Centers for Disease Control and Prevention (CDC) standard [19]. DNA fragments were digested by *Xbal* and separated by a CHEF-DRIII Pulsed-Field Electrophoresis System. The gel was stained with ethidium bromide and documented using a ChemiDoc^TM^ XRS+ (Bio-Rad, Hercules, California, USA). The dendrogram was produced using band clustering with a dice coefficient similarity index of 1% optimization and 1% tolerance by the unweighted pair group method with arithmetic means (UPGMA) by BioNumerics version 7.6 (Applied Maths, Keistraat, Sint-Martens-Latem, Belgium).

To prepare the tests for FTIR, the PFGE confirmed isolates were performed as follows: The *Salmonella* isolates were cultured in nutrient broth for 24 h at 28 °C, and then the culture was centrifuged at 6000 rpm for 5 min. Cell pellets were washed with 0.85% NaCl, then re-centrifuged. The culture was dissolved in sterile distilled water and deposited into a BaF_2_ window. The prepared test windows were scanned by SR-FTIR microspectroscopy at the Synchrotron Light Research Institute (Public Organization), Thailand. The infrared spectra were collected using a Bruker IR spectrometer (VERTEX70) coupled to an IR microscope (Hyperion 2000 IR microscope coupled with VERTEX70 spectrometer, Bruker Optics, Ettlingen, Germany). Spectrum acquisition from 4000–600 cm^−1^ with transmission mode were collected by the OPUS 7.5 software (Bruker Optics Inc., Billerica, MA, USA) with a 36× objective lens, a background scan time of 64 scans and a resolution of 4 cm^−1^. Principal component analysis (PCA) and soft independent modelling of class analogy (SIMCA) were analyzed by the Unscrambler X 10.5 software (CAMO Software Inc., Newfoundland, Canada). All spectra of each test were transformed by second derivative and vector normalization in the range from 3000–2800 cm^−1^ and 1750–950 cm^−1^, according to principal component 1 (most principal component contributing for clustering, PC1). The PCA was used to classify based on the differentiation of the biochemical components in the whole cells from each strain. The SIMCA was used for the classification of spectrum data, which requires a training data set to identify the class membership. The training data set was from the spectra of the PFGE-matched isolates from dogs. The test data were the spectra of the PFGE-matched isolates from humans. The class memberships were shown by the percentage of correctly identified spectra data by SIMCA analysis.

## 3. Results 

### 3.1. The Prevalence and AMR of Salmonella from Pet Dogs and Humans in Khon Kaen, Thailand

The prevalence of *Salmonella* among dogs and humans was 12.86% (18/140) and 17.86% (25/140), respectively. Of all the positive samples, 13 *Salmonella* serotypes from 18 pet dog isolates were identified, and 13 serotypes from 25 human isolates were identified. The dominant serotypes (more than 10%) detected from dogs were *S. enterica* serotype Stanley (16.67%), *S. enterica* serotype Hvittingfoss (16.67%), and *S. enterica* serotype I 1,4,[5],12:i:- (11.20%). The dominant serotypes identified from humans were *S. enterica* serotype Stanley (24.00%), *S. enterica* serotype Weltevreden (16.00%), and *S. enterica* serotype I 1,4,[5],12:i:- (12.00%) (Figure 1).

Antimicrobial susceptibility testing of *Salmonella* among dogs and human caregivers determined antimicrobial resistance to: ampicillin (AMP) (44.44% and 64.00%), chloramphenicol (CHL) (16.67% and 24.00%), ciprofloxacin (CIP) (5.56% and 12.00%), nalidixic acid (NA) (5.56% and 12.00%), streptomycin (STR) (22.22% and 32.00%), sulfamethoxazole–trimethoprim (SXT) (16.67% and 24.00%) and tetracycline (TET) (38.89% and 64.00%), respectively. The total AMR containing *Salmonella* isolates from pet dogs was 44.44% (8/18), and 64.00 % (16/25) in human caregivers.

### 3.2. The Epidemiological Relatedness of Salmonella Infection among Paired Dogs and Humans

The *Salmonella* species were identified from 43 individuals of 39 households (dogs only (*n* = 14), humans only (*n* = 21), both dogs (*n* = 4) and humans (*n* = 4)), whereas 101 households were *Salmonella* negative. The tally of *Salmonella* infected dogs and humans, from paired data, is shown in Table 1. The relationship between pet dogs and their human caregivers for the occurrence of *Salmonella* infection was not statistically different (*p* = 0.310, *n* = 140) when analyzed by McNemar’s test. Thus, the *Salmonella* presence between pet dogs and humans was similar to each other.

Of the serotypes, nine serotypes (including *S. enterica*: Altona, Bredeney, Derby, Hvittingfoss, type I 1,4,[5],12:i:-, Kedougou, Krefeld, Stanley, and Weltevreden) were shared between the isolates of pet dogs and humans. The mutual serotype isolates in pet dogs were 77.78% (14/18), and 84.00% (21/25) in humans (Figure 2). The serotypes of the *Salmonella* isolates from pet dogs and humans were not significantly different (*p* > 0.05), indicating the serotypes among pet dogs are quite similar to humans.

For the prevalence of AMR patterns, the AMR patterns of *Salmonella* isolates from the paired samples were shown in Table 2. The isolates of AMR patterns between the pet dog source and the human source were statistically analyzed using Pearson’s chi-squared test. The resistance patterns of *Salmonella* isolates were not significantly different between human and pet dog isolates (*p* > 0.05). Thus, this demonstrates that the AMR patterns of the isolates from the two populations among pet dogs and human caregivers were similar.

After the determination of homologous *Salmonella* isolates among paired individuals (serotype screening, and confirmed by PFGE and SR-FTIR), two serotype-matched pairs, H121 and D121, and H86 and D86, were found to be homologous isolates (Figure 3). The 18 pet dog isolates and 25 human isolates were screened according to the serotype, shown by Venn diagram (Figure 3A). The two-paired serotype matched isolates (H121 and D121, and H86 and D86) were at the 100% similarity level, which was confirmed by the PFGE method (Figure 3B). The PFGE-matched isolates were further confirmed by SR-FTIR and analyzed using PCA and SIMCA. All spectra of each test were transformed by second derivative and vector normalization in the range from 3000–2800 cm^−1^ and 1750–950 cm^−1^, according to principal component 1 (PC1). The loading PC1 (most component contributing for clustering) was introduced to adjust the best-distinguished range of spectra for further analysis (PCA and SIMCA), which explained the distinguished peak position. The positive loading PC1 showed that 1739, 1650, 1540, and 1240 cm^−1^ were the distinguishing peaks from the negative side of the PCA score plot (H121 and D121 were grouped into the left side of PCA score plot). The negative side of the loading plot separated D86 and H86, and corresponded to distinguished peaks at 977, 1664, and 1641 cm^−1^. Therefore, the distinguished range of spectra was set at 3000–2800 cm^−1^ and 1750–950 cm^−1^ (Figure 3C). The PCA score plot separated the spectra of isolate pair H121 and D121 from H86 and D86 (Figure 3D). The class memberships analyzed by SIMCA were shown by the percentage of correctly identified spectra data. The percentage of correctly identified spectra of H121 by model-D121 was 96.72%, whereas the percentage of model-D86 was 0.82%. This indicated that the class membership of isolate H121 belongs to model-D121. The percentage of correctly identified spectra of H86 by model-D86 was 95.92%, whereas the percentage of model-D121 was 30.61%. This indicated the class membership of isolate H86 belongs to model-D86 (Figure 3E). The results of PCA and SIMCA revealed that the spectra of isolates H121 and D121 were the same, as for isolates H86 and D86 (Figure 3D,E). The homologous *Salmonella* isolate pairs indicate that the corresponding pet dog and human pairs were infected with the same *Salmonella,* either from passage between each other or the *Salmonella* was obtained individually from the same source (i.e., the same food-borne source or same infection source). The prevalence of homologous *Salmonella* pairs from the total *Salmonella*-present households was 5.13% (2/39).

## 4. Discussion

The prevalence of *Salmonella* in pet dogs determined herein was 12.86%, which was similar to that reported in a previous investigation in Thailand (13.2%) [9], and in Ethiopia (11.7%) [20], while it was much higher than that reported in the USA (2.5%) [21]. The *Salmonella* prevalence of 17.86% in humans was higher than the 4.7% reported in Thailand [13]. As for the most predominant serotypes, *S. enterica* Stanley was one of the most common serotypes among dogs reported in Thailand [13]. Moreover, it was one of the ten most common *S. enterica* serotypes from Thai patients, along with Weltevreden and I 1,4,[5],12:i:- [22]. In Northeast Thailand, the *S. enterica* serotype Hvittingfoss was one of the top ten most common *Salmonella* serotypes detected [22]. Thus, the prevalence of *Salmonella* from pet dogs and humans in this study is consistent with other reports, and the serotypes detected herein are among the predominant serotypes reported in Thailand.

The prevalence of antimicrobial resistant isolates from pet dogs in this study (44.44%) is lower than the reports from dogs in Addis Ababa, Ethiopia (90.5%) [20], while it is similar to the report of dogs from Khon Kaen, Thailand (54.8%) [9]. The prevalence of antimicrobial resistant *Salmonella* isolates from human caregivers in this study (64.00%) was lower than human patients of 100% in Khon Kaen, Thailand [23]. The most prevalent antimicrobial resistant agents included ampicillin and tetracycline. The susceptibility patterns of this study correspond to other reports in Thailand for ampicillin, streptomycin, and tetracycline [24,25]. The ratios of antimicrobial resistant *Salmonella* isolates from pet dogs were lower than humans, revealing a situation of more antimicrobial resistance among humans than pet dogs. However, the presence of antimicrobial resistance among the *Salmonella* isolates from both dogs and humans revealed a serious situation of antimicrobial resistance in Khon Kaen, Thailand.

For the epidemiological relatedness results of *Salmonella* infection among paired dogs and humans, (1) the relationship between pet dogs and their human caregivers for the occurrence of *Salmonella* infection was not statistically different (*p* = 0.310, *n* = 140) when analyzed by McNemar’s test; (2) the *Salmonella* serotypes between pet dogs and humans were not significantly different (*p* > 0.05); and (3) the resistance patterns of *Salmonella* isolates were not significantly different between human and pet dog isolates (*p* > 0.05). This reveals that the presence of *Salmonella*, the AMR patterns, and the serotypes from pet dogs and humans were quite similar. However, the number of homologous *Salmonella* isolate pairs (the cases that were infected between each other or from similar sources) among the total *Salmonella*-present households was only 5.13%, indicating that the assumption of high risk close contact infection between pet dogs and their caregivers [7,8,9] was doubtful. The data suggests that close contact between pet dogs and humans was not the main cause of *Salmonella* infection among humans in Khon Kaen, Thailand (the true rate of close contact should be equal to or lower than the rate of the homologous *Salmonella* isolate pairs). This conclusion is consistent with the research that household pet dogs as a source of *Salmonella* was less important than other exposures [6]. The reason for the similar *Salmonella* presence, AMR patterns, and serotypes with a low proportion of homologous *Salmonella* isolate pairs, is possibly due to the similar living environments between pet dogs and human caregivers but a low risk of infection passage between the two populations. To obtain credible information on the risk of pet dogs as a source AMR *Salmonella,* more research on the evaluation of other areas are needed as living styles vary by region.

In this study, paired sampling was applied in the investigation of the epidemiological association of AMR *Salmonella* between dogs and their caregivers. The idea came from noting the role of AMR bacteria transmission among animals and humans, which may play an important part in the AMR problem, in which blocked critical points may reduce AMR bacteria transmission [1,2]. However, there is limited data on the transmissibility of AMR bacteria among humans. To explore the risk of the critical points of AMR bacteria transmission, the epidemiological association of AMR *Salmonella* infections between pets and their caregivers (i.e., the close contact way for AMR *Salmonella* spread) in Khon Kaen was investigated in this work. This paired sample investigation seems like a variant of the cross-sectional study. The variables of a cross-sectional study are measured at a single time. It comprises the measurement of exposure and/or outcome. Traditional cross-sectional studies benefit by assessing the burden of disease in a population, examining the trends of disease, and comparing the disease prevalence of different populations [26]. The paired sample investigation was carried out on two matched populations. It provided more information on the differences and similarities compared to traditional cross-sectional studies. This work benefited from the use of paired samples, in which the low proportion of homologous *Salmonella* isolate pairs revealed that close contact between dogs and caregivers was not a major factor of human AMR *Salmonella* infections in Khon Kaen, Thailand, which proved to be a counterexample for the assumption of the high risk of close contact infection between dogs and their caregivers [7,8,9]. Lastly, this work was limited in: (1) the ability to confirm the *Salmonella* transmission route among the study participants (e.g., the homologous *Salmonella* isolate pairs may have been due to transmission of infection between the dog and caregivers, or the infection may have been due to transmission to each individual by the same source and therefore not from each other); (2) the low quantity of pet dog source AMR *Salmonella* evaluations from other regions; (3) not having available symptomatic pet dogs and humans with diarrhea which could have influenced the results.; (4) determining a higher confidence level of the *Salmonella* prevalence among the study participants due to the limitations of bacterial cultures.

## 5. Conclusions

In this study, the presence of similar *Salmonella* isolates, AMR patterns, and serotypes with a low proportion of homologous *Salmonella* isolate pairs was found, possibly due to similar living environments among pet dogs and human caregivers. Moreover, it was determined that a low proportion of AMR *Salmonella* infections between pet dogs and humans out of the total *Salmonella*-present households was related to each other. Thus, close contact between pet dogs and their human caregivers might not be the major factor for *Salmonella* infections or the spread of AMR *Salmonella* in Khon Kaen, Thailand. More *Salmonella* research is needed to include studies with larger sample sizes and higher *Salmonella* detection methods to determine the presence, prevalence, distribution and methods of transmission.

## Figures and Tables

**Figure 1 antibiotics-09-00477-f001:**
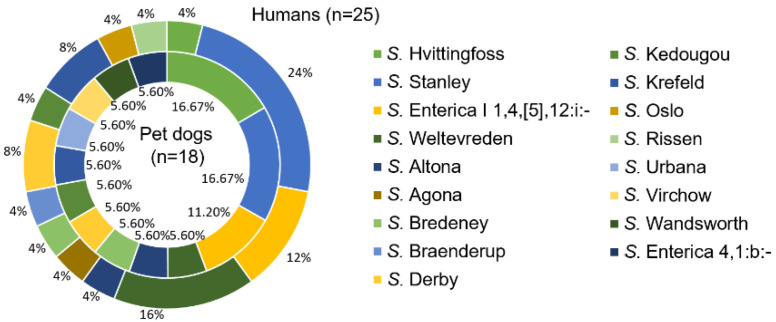
The serotypes of *Salmonella* from pet dogs and humans in Khon Kaen, Thailand. Note: thirteen serotypes from 18 *Salmonella* isolates were isolated from pet dogs, and 13 serotypes with 25 *Salmonella* isolates were from humans. The inner circle denotes the serotypes from pet dog source isolates. The outer circle denotes serotypes from human source isolates. The colors of the two circles denote different *Salmonella* serotypes.

**Figure 2 antibiotics-09-00477-f002:**
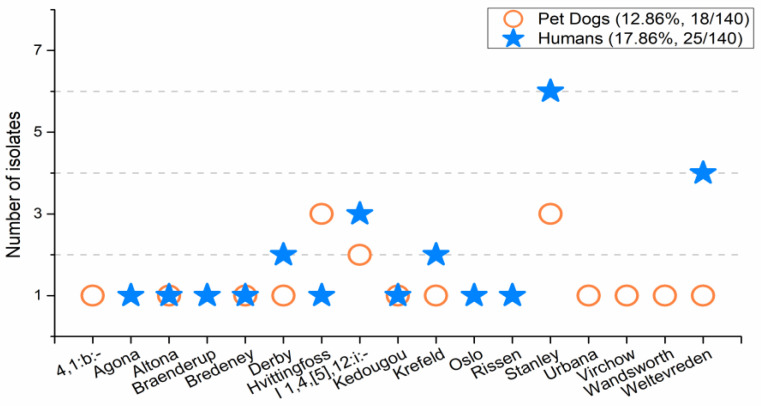
The serotypes of *Salmonella* from paired pet dogs and humans in Khon Kaen, Thailand. Note: the serotypes of pet dog fecal samples were quite similar to the serotypes from humans. Thirteen *Salmonella* serotypes were identified from pet dogs, and thirteen serotypes were from humans.

**Figure 3 antibiotics-09-00477-f003:**
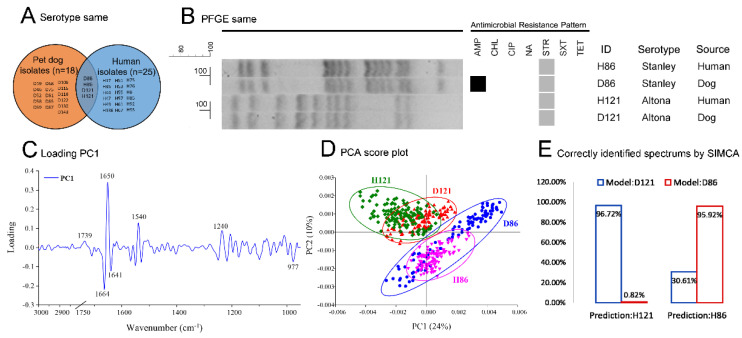
The determination of homologous *Salmonella* isolate pairs (■ = resistance, ■ = intermediate). Note: (**A**) serotype screening result, shown by Venn diagram. Two pair isolates were serotype matched (H121 and D121, and H86 and D86). The code behind ‘H’ or ‘D’ denotes the household number (i.e., H86 and D86 are from the same household). (**B**) The isolates H121 and D121, and H86 and D86 exhibited a 100% similarity level as confirmed by the pulsed field gel electrophoresis (PFGE) method. (**C**) Loading principal component 1 (PC1) result, the most principal component contributes for clustering. (**D**) Principal component analysis (PCA) result. (**E**) Soft independent modelling of class analogy (SIMCA) result. (**C**–**E**) The PFGE matched isolates were further confirmed by Synchrotron Fourier transform infrared spectroscopy (SR-FTIR) and analyzed using PCA and SIMCA. The results of the PCA and SIMCA revealed that the spectra of isolates H121 and D121 were the same, as for H86 and D86.

**Table 1 antibiotics-09-00477-t001:** Tally of *Salmonella*-positive samples from pet dogs and their caregivers.

		Human Caregivers	
		Positive	Negative
**Pet dogs**	Positive	4	14
	Negative	21	101

Note: the tally of *Salmonella*-positive samples was sorted to assess the dependence of these paired samples using McNemar’s test. The statistical difference of McNemar’s test (*p* < 0.05) reveals that the presence of *Salmonella* among paired samples was independent of each other.

**Table 2 antibiotics-09-00477-t002:** Antimicrobial resistance (AMR) patterns of *Salmonella* from pet dogs and their caregivers.

Antimicrobial Resistance Patterns	Isolate Numbers from Pet Dogs	Isolate Numbers from Humans
AMP	1	0
AMP-TET	2	4
AMP-STR	0	1
CHL-TET	0	1
AMP-STR-TET	2	6
AMP-CHL-SXT-TET	1	1
AMP-CHL-STR-SXT-TET	1	1
AMP-CHL-CIP-NA-SXT-TET	0	2
AMP-CHL-CIP-NA-STR-SXT-TET	1	1
Sensitive to all antimicrobials	10	8

Note: AMP: ampicillin, CHL: chloramphenicol, CIP: ciprofloxacin, NA: nalidixic acid, STR: streptomycin, SXT: sulfamethoxazole–trimethoprim, TET: tetracycline. The AMR patterns of the isolates from the two populations were not significantly different (*p* > 0.05) using Pearson’s chi-squared test. The antimicrobial-sensitive isolates were not considered in the statistical analysis.

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
