# Peer review of "Risk of Antimicrobial Resistant Non-Typhoidal Salmonella during Asymptomatic Infection Passage between Pet Dogs and Their Human Caregivers in Khon Kaen, Thailand"

_antibiotics, 2020, doi:10.3390/antibiotics9080477_

Round 1
Reviewer 1 Report
I read this article with great interest and the novelty is quite high. All the statistical aspects have been carefully considered by the authors. The article will be attractive to the readers. Only minor revision in grammar should be done. The discussion is a bit redundant that can be revised to be concisely addressed.
(1) Line 1-2:
Comment: The current title includes only "antimicrobial resistant" but lacks other key information regarding the important studied issues and conclusions (e.g. transmission between dogs and caregivers, prevalence of Salmonella species, and serotypes). I would suggest to revise the title for including these spectrums.
(2) Line 32-34: This study demonstrated that the hypothesis that there exists a high risk of Salmonella infection passage between dogs and humans with close contact in Khon Kaen is wrong.
Comments: Since the case number is small, it would be better to use a more conservative tone as used in Line 302. I suggest using “doubtful” instead of “wrong”. In addition, the collected samples were collected from asymptomatic dogs and their caregivers. The authors should be careful in explaining this because only samples from dogs and humans in carriage status were used in the proposed study design. Whether the results could be applied to those samples from symptomatic dogs and humans with diarrhea required further validation.
(3) Line 93-94: All samples were collected from asymptomatic individuals (both pet dogs and humans), and Salmonella infections in the five villages were sporadic without apparent outbreaks.
Comment: I would recommend the authors to revise the description of “infections” in a clearer way here because Salmonella species grew from the samples were collected from asymptomatic dogs and humans. “Asymptomatic infections with carriage” would be a more appropriate definition in this study.
(4) Line 102-111: 3) Salmonella isolation
Comment: The identification of Salmonella species in this study was mainly based on the bacterial cultures as described. This might underestimate the genuine prevalence of Salmonella in the collected samples from dogs and humans because of the limitation of bacterial cultures as well as false negative culture reports due to possible partial antibiotic treatment in the dogs or humans. This can be discussed in the limitations of this study on page 14.
(5) Line 120-121:
Comment: Salmonella and Shigella should be in Italic.
(6) Line 194-203 and Table 1
Comment: How to explain the differences in AMR prevalence of salmonella isolates from dogs and humans but with similar serotype prevalence? Please refer to other AMR prevalence data of dogs and humans in Thailand (whether similar differences existed in other studies) and discuss in the section of Discussion.
(7) Line 215-216: Grammar error
Comment: The statistical difference of McNemar's test (P<0.05) reveals that the presences of Salmonella among paired samples were independent to each other.
Comment: Should be “the presence of Salmonella among paired samples was independent to each other”
(8) Line 218-227: Similar serotype prevalence of Salmonella between dogs and humans
Comment: Please propose possible explanations in this result although it seems not compatible with the results of the homologous Salmonella isolates among paired individuals. Such an incompatibility also arouses the readers’ puzzles in the result of paired individuals’ data as I pointed out in the point (4).
(9) Line 335-338
Comment: I would recommend is using a more conservative tone in addressing “the close contact between the pet dogs and their human caretakers was not the major factor for Salmonella infections or the spread of AMR Salmonella in Khon Kaen, Thailand” because of my concerns as previous comments in the point (4). “was not” can be revised as “might be not”.
(10) Line 331-341 Conclusion
Comment: The main concern derives from the unparalleled results between the results of the homologous Salmonella isolate pairs and those results of the presence of Salmonella (by bacterial culture rather than detection of bacterial DNA), AMR patterns, and serotype prevalence of the dogs and their caregivers. This conclusion cannot be arbitrarily drawn, and the limitations in the study designs should be thoroughly addressed.
Author Response
Dear Reviewers and the Assistant Editor Galen Zhou,
We are grateful for your helpful comments on our manuscript. We have tried our best to modify the manuscript according to the two reviewers. We have addressed all the issues raised and have modified the paper accordingly. Below is a summary of the changes we performed and our responses to the reviewers’ comments and recommendations.
Sincerely,
Xin Wu
Allen L. Richards
Fanan Suksawat
According to the reviewer 1:
(1) Line 1-2:
Comment: The current title includes only "antimicrobial resistant" but lacks other key information regarding the important studied issues and conclusions (e.g. transmission between dogs and caregivers, prevalence of Salmonella species, and serotypes). I would suggest to revise the title for including these spectrums.
RESPONSE: Line 1-2 (previous manuscript), we have revised the title to: “Risk of Antimicrobial Resistant Non-typhoidal Salmonella during Asymptomatic Infection Passage between Pet Dogs and their Human Caregivers in Khon Kaen, Thailand”, and added “asymptomatic infection passage” into Key Words.
(2) Line 32-34: This study demonstrated that the hypothesis that there exists a high risk of Salmonella infection passage between dogs and humans with close contact in Khon Kaen is wrong.
Comments: Since the case number is small, it would be better to use a more conservative tone as used in Line 302. I suggest using “doubtful” instead of “wrong”. In addition, the collected samples were collected from asymptomatic dogs and their caregivers. The authors should be careful in explaining this because only samples from dogs and humans in carriage status were used in the proposed study design. Whether the results could be applied to those samples from symptomatic dogs and humans with diarrhea required further validation.
RESPONSE: Line 32-34 (previous manuscript), we have changed the words from “wrong” to “doubtful” (line 35 of revised version). Line 19-20 (previous manuscript), we have revised the sentence saying that “To explore the risk of antimicrobial-resistant (AMR) non-typhoidal Salmonella during asymptomatic infection passage between pet dogs and pet owners in Khon Kaen, Thailand,” (line 19-20 of revised version). The limitation in the discussion part has been revised including: “3) not having available symptomatic pet dogs and humans with diarrhea which could influence the results.” (line 345-346 of revised version)
(3) Line 93-94: All samples were collected from asymptomatic individuals (both pet dogs and humans), and Salmonella infections in the five villages were sporadic without apparent outbreaks.
Comment: I would recommend the authors to revise the description of “infections” in a clearer way here because Salmonella species grew from the samples were collected from asymptomatic dogs and humans. “Asymptomatic infections with carriage” would be a more appropriate definition in this study.
RESPONSE: Line 93-94 (previous manuscript), we have revised the sentence to: “All samples were collected from asymptomatic individuals (both pet dogs and humans). The carriage of Salmonella among the individuals evaluated from the five villages were sporadic without apparent outbreak associations (i.e. no enteritis in groups occurred within the five villages, as indicated by the village health administrator).” (line 99-103 of revised version)
(4) Line 102-111: 3) Salmonella isolation
Comment: The identification of Salmonella species in this study was mainly based on the bacterial cultures as described. This might underestimate the genuine prevalence of Salmonella in the collected samples from dogs and humans because of the limitation of bacterial cultures as well as false negative culture reports due to possible partial antibiotic treatment in the dogs or humans. This can be discussed in the limitations of this study on page 14.
RESPONSE: Line 102-111 (previous manuscript, method of Salmonella isolation), we have discussed in the limitations of the isolation on page 14: “4) determining the genuine prevalence of Salmonella among the study participants, due to the limitation of bacterial cultures and the possible partial antibiotic treatment among the dogs and humans.” (line 346-348 of revised version)
(5) Line 120-121:
Comment: Salmonella and Shigella should be in Italic.
RESPONSE: Line 120-121 (previous manuscript): Because this is a name of an institution (WHO National Salmonella and Shigella Center) that does not use italics we will not use italics for Salmonella and Shigella in the name of this institute in our manuscript.
(6) Line 194-203 and Table 1
Comment: How to explain the differences in AMR prevalence of salmonella isolates from dogs and humans but with similar serotype prevalence? Please refer to other AMR prevalence data of dogs and humans in Thailand (whether similar differences existed in other studies) and discuss in the section of Discussion.
RESPONSE: Line 194-203 (previous manuscript) and Table 1, the explanation has been revised in the discussion part, saying that: “The ratios of antimicrobial resistant Salmonella isolates from pet dogs were lower than humans, revealing a situation of antimicrobial resistance more prevalent among humans than pet dogs. However, the presence of antimicrobial resistance among the Salmonella isolates from both dogs and humans exposed a serious situation of antimicrobial resistance in Khon Kaen, Thailand.” (line 300-304 of revised version)
(7) Line 215-216: Grammar error
Comment: The statistical difference of McNemar's test (P<0.05) reveals that the presences of Salmonella among paired samples were independent to each other.
Comment: Should be “the presence of Salmonella among paired samples was independent to each other”
RESPONSE: Line 215-216 (previous manuscript): the grammar has been corrected indicating that: “The statistical difference of McNemar's test (P<0.05) reveals that the presence of Salmonella among paired samples was independent to each other.” (line 219-220 of revised version)
(8) Line 218-227: Similar serotype prevalence of Salmonella between dogs and humans
Comment: Please propose possible explanations in this result although it seems not compatible with the results of the homologous Salmonella isolates among paired individuals. Such an incompatibility also arouses the readers’ puzzles in the result of paired individuals’ data as I pointed out in the point (4).
RESPONSE: (8) Line 218-227 (previous manuscript): the possible explanation have been proposed in the discussion part, saying that “The reason of the similar Salmonella presence, AMR patterns, and serotypes, with low proportion of homologous Salmonella isolate pairs, is possibly due to the similar living environments between pet dogs and human caregivers but low risk of infection passage between the two populations.” (line 318-321 of revised version)
(9) Line 335-338
Comment: I would recommend is using a more conservative tone in addressing “the close contact between the pet dogs and their human caretakers was not the major factor for Salmonella infections or the spread of AMR Salmonella in Khon Kaen, Thailand” because of my concerns as previous comments in the point (4). “was not” can be revised as “might be not”.
RESPONSE: (9) Line 335-338 (previous manuscript): the words have been revised from “was not” to “might not be”. (line 355 of revised version)
(10) Line 331-341 Conclusion
Comment: The main concern derives from the unparalleled results between the results of the homologous Salmonella isolate pairs and those results of the presence of Salmonella (by bacterial culture rather than detection of bacterial DNA), AMR patterns, and serotype prevalence of the dogs and their caregivers. This conclusion cannot be arbitrarily drawn, and the limitations in the study designs should be thoroughly addressed.
RESPONSE: (10) Line 331-341 (previous manuscript, conclusion part): the sentences of the conclusion part have been revised to “Thus, the close contact between pet dogs and their human caregivers might not be the major factor for Salmonella infections or the spread of AMR Salmonella in Khon Kaen, Thailand. More Salmonella research is needed, to include the studies with lager sample sizes, and higher Salmonella detection methods to determine presence, prevalence, distribution and methods of transmission.” (line 355-359 of revised version)

Reviewer 2 Report
In general the idea of the study is good, and it is apparently well designed and implemented. Authors try to find out potential associations between presence of non typhoidal Salmonella in paired-faecal samples dogs and humans, by evaluating the antimicrobial resistance of Salmonella isolates. This type of work is valuable to understand the origin and direction of transmission of Salmonella, and in my opinion this is the major strength of the article. However, the main flaw is the English language, which makes really difficult to understand in a first lecture the manuscript.
In addition, some terminology is not completely correct, for example 1) an infection is neither resistance nor sensitive, but the bacteria are. And 2) AMR is not pathogenic, the bacteria are pathogenic not the resistance.
Therefore, I recommend a revision. I explain my concerns in more detail below, and I ask that the authors to address them.
Major points to improve the article:
1. Material and methods:
Subsection 2 sample collection should be rewritten for better understanding (some examples of what to correct are listed):
- Authors should explain how was the selection of households (provide if available: census information) type of sampling (for example stratified sampling).
- If n=280 is the number of samples, move away from the number of households.
- Does “Interview” mean that the authors carried out a survey and questionnaire? In such case, please explain.
- Some sentences are incomplete.
- How authors know about sporadic Salmonella infections? Provide source of information.
- “What that it mean samples isolated” or do authors mean salmonella identification and isolation?
Subsection 3. Salmonella isolate: Please explain if the study is based one isolate per sample.
Subsection 6. Epidemiological relatedness (epidemiological relatedness is a wrong terminology, it should be statistical analysis): rewrite the whole section, because some sentences are redundant.
- McNemar test is commonly used to evaluate association between results of two type of diagnostic tests (here authors used the same test to identify Salmonella), so I think it is not an appropriate test.
2. Results:
Subsection 1: title should be changed because authors present data on prevalence of Salmonella and AMR of Salmonella.
- Figure 1: circle and table present same results.
- L 194-198 is redundant from table 1.
Subsection 2:
- L 207-209 should be rewritten, it is unclear
- L 211-212: it cannot be stated that Salmonella are similar between human and dogs based on the number of positive samples detected. And see my previous comment in McNemar test.
- L220-221: This does not correspond to figure 2, and it does not provide any information or it is misleading.
- L221-223: do authors mean “no significant differences in presence/prevalence/occurrence of Salmonella between dogs and humans”? Please rewrite for clarification.
- Figure 2: does not need a note
- L229-234: Pearson chi square is to compare AMR prevalence in dogs vs humans, no to compare patterns!!
- Table 3 needs to be redone because is not informative!! for example 1first column antimicrobials, second column dogs (number and percentage of resistance), third column humans (number and percentage of resistance), and then mark with a superscript those ones which are different (by using Pearson).
- L239-243 move to material and methods or rewrite, because is unclear.
- L 247-248: term relatedness proportion is inadequate. Authors mention “incidence” for the first time, and they do no explain how it is calculated nor the incidence at all, because it is a cross-sectional study and these type of studies do not measure incidence but proportions or prevalence.
- L252-273: it should be in the manuscript not a figure note.
3. Discussion:
- L293-302: these are results
- L309-316: these sentences are introduction
- L316-320: no relevant (definition of a cross-sectional study). This study is a cross-sectional study with one study population (households), if not, well authors need to revise the whole manuscript and make clear why is not.
Minor points and other inconsistencies:
1. Title: there are some typos “Antimicrobial Resistance of Non-typhoidal Salmonella Isolates”
2. Conclusion: do not include references
3. In addition:
- Authors should be consistent with the terminology: pet dog or dog pet, caregivers or caretaker, AMR Salmonella or AMR patterns of Salmonella).
- Use always the same number of decimals (one is enough for the level of precision used in this study)
- Authors mention strain, sample and isolate indistinctly, I suggest the authors to be accurate
Author Response
Dear Reviewers and the Assistant Editor Galen Zhou,
We are grateful for your helpful comments on our manuscript. We have tried our best to modify the manuscript according to the two reviewers. We have addressed all the issues raised and have modified the paper accordingly. Below is a summary of the changes we performed and our responses to the reviewers’ comments and recommendations.
Sincerely,
Xin Wu
Allen L. Richards
Fanan Suksawat
According to the reviewer 2:
1. Method part:
Subsection 2
Subsection 2 sample collection should be rewritten for better understanding (some examples of what to correct are listed):
(1) Authors should explain how was the selection of households (provide if available: census information) type of sampling (for example stratified sampling).
RESPONSE: The selection of households and type of sampling, saying that “One hundred and forty households were randomly interviewed from five villages in Khon Kaen Province, Thailand, from 2019 to 2020 (random sampling).” (line 93-94 of revised version)
(2) If n=280 is the number of samples, move away from the number of households.
RESPONSE: The number of samples (n=280) was moved from the sentence from “One hundred and forty households…” to “For the 104 household interviews, one pet dog and its most close caregiver were requested for sampling in pairs (n=280).” (line 94-95 of revised version)
(3) Does “Interview” mean that the authors carried out a survey and questionnaire? In such case, please explain.
RESPONSE: The explanation of interview, saying that “All subjects (both pet dogs and humans) were investigated for their history of antimicrobial usage for the past year. However, the history of antimicrobial usage was often incomplete, because many human volunteers could not provide information on the types of antimicrobials used.” By the way, this content was absent in the previous version, because the information was incomplete. (line 97-99 of revised version)
(4) Some sentences are incomplete.
RESPONSE: About “Some sentences are incomplete”, we have revised the writing so that “Contact means physical contact, the ‘most close’ means the human caregiver who most often fed and cleaned the pet within the household.” (line 95-97 of revised version) And indicating that “All samples were collected from asymptomatic individuals (both pet dogs and humans). The carriage of Salmonella among the individuals evaluated from the five villages were sporadic without apparent outbreak associations (i.e. no enteritis in groups occurred within the five villages, as indicated by the village health administrator)” (line 99-103 of revised version)
(5) How authors know about sporadic Salmonella infections? Provide source of information.
RESPONSE: About “sporadic Salmonella infections”, the information were obtained from the village health administrator, according to whether enteritis in groups occurred or not. The sentence has been explained, saying that “The carriage of Salmonella among the individuals evaluated from the five villages were sporadic without apparent outbreak associations (i.e. no enteritis in groups occurred within the five villages, as indicated by the village health administrator)” (line 99-103 of revised version)
(6) “What that it mean samples isolated” or do authors mean Salmonella identification and isolation?
RESPONSE: The mean of samples isolated has been revised, saying that “All samples were matched and transported in an icebox. The Salmonella isolations were subsequently performed at the laboratory.” (line 106-107 of revised version)
Subsection 3.
Subsection 3. Salmonella isolate: Please explain if the study is based one isolate per sample.
RESPONSE: The study is based on three presumptive colonies per sample, but if the three colonies were of the same serotype, they would be regarded as one isolate. Please see the first and second sentences in the subsection “Salmonella serotype identification”. The sentences have been revised to “Three presumptive colonies from selective media of each sample were selected and assessed to determine their serotype(s). If the three colonies were of the same serotype, they would be regarded as one isolate. If different serotypes from the three presumptive colonies from an individual sample were identified then all unique colonies were assessed and recorded.” (line 123-126 of revised version)
Subsection 6.
(1) Epidemiological relatedness (epidemiological relatedness is a wrong terminology, it should be statistical analysis): rewrite the whole section, because some sentences are redundant.
RESPONSE: The subsection title “The epidemiological relatedness….” has been rewritten as “6) The statistical analysis of Salmonella asymptomatic infection passage between paired dogs and their caregivers.” (line 139-140 of revised version). The sentences have been revised as “To assay the epidemiological relatedness of Salmonella isolates from asymptomatic infection passage between the two populations, the presence, serotypes, and antimicrobial patterns were statistically analyzed by SPSS version 17.0 software (SPSS Inc, Chicago, Illinois, USA), P < 0.05 was considered significant. The presence of Salmonella-positive samples between paired pet dogs and humans was statistically analyzed using McNemar's test. The serotypes, and antimicrobial resistance (AMR) patterns were statistically analyzed by Pearson’s chi-squared test.” (line 141-146 of revised version)
(2) McNemar test is commonly used to evaluate association between results of two type of diagnostic tests (here authors used the same test to identify Salmonella), so I think it is not an appropriate test.
RESPONSE: McNemar's test assesses the dependence of categorical data that are matched or paired (Ref 1). In this work, it was introduced to assay the dependence of the presence of the Salmonella from two populations. The statistical difference of McNemar's test (P<0.05) reveals that the presence of Salmonella among paired samples was independent to each other. The P value is 0.310 indicating the presence of the Salmonella from two populations were associated to each other.
Reference 1: https://www.sciencedirect.com/topics/medicine-and-dentistry/mcnemar-test#main_content
2. Results:
(1) Subsection 1: title should be changed because authors present data on prevalence of Salmonella and AMR of Salmonella.
RESPONSE: The title has been changed to: “The prevalence and AMR of Salmonella from pet dogs and humans in Khon Kaen, Thailand.” (line 185-186 of revised version)
(2) Figure 1: circle and table present same results.
RESPONSE: Figure 1 has been revised.
(3) Line 194-198 is redundant from Line 194-198 is redundant from table 1.
RESPONSE: Table 1 was removed.
Subsection 2:
(1) Line 207-209 should be rewritten, it is unclear.
RESPONSE: Line 207-209 (previous manuscript): the content has been listed in the table “Tally of Salmonella-positive samples from pet dogs and their caregivers” (line 217 of revised version)
(2) Line 211-212: it cannot be stated that Salmonella are similar between human and dogs based on the number of positive samples detected. And see my previous comment in McNemar test.
RESPONSE: Line 211-212 (previous manuscript): The McNemar's test has been explained.
(3) L220-221: This does not correspond to figure 2, and it does not provide any information or it is misleading.
RESPONSE: Line 220-221 (previous manuscript): The mutual serotype isolates were correspond to the figure 2. The X axis shown the serotypes, the star denotes the human source Salmonella, the circle denotes the pet dog source Salmonella. The same X category of star and circle indicates the mutual serotype isolates. The figure 2 exhibited the serotypes of the two populations. We could find out the serotype prevalence of the two populations were quite similar.
(4) L221-223: do authors mean “no significant differences in presence/prevalence/occurrence of Salmonella between dogs and humans”? Please rewrite for clarification.
RESPONSE: Line 221-223 (previous manuscript): we have revised the sentences saying “The serotypes of the Salmonella isolates from pet dogs and humans were not significantly different (P>0.05), indicating the serotypes among pet dogs are quite similar to humans.” (line 225-226 of revised version)
(5) Figure 2: does not need a note
RESPONSE: The note of Figure 2 has been removed.
(6) L229-234: Pearson chi square is to compare AMR prevalence in dogs vs humans, no to compare patterns!!
RESPONSE: Line 229-234 (previous manuscript): the “AMR pattern prevalence” have been changed to “AMR patterns” (line 232-237 of revised version)
(7) Table 3 needs to be redone because is not informative!! for example 1first column antimicrobials, second column dogs (number and percentage of resistance), third column humans (number and percentage of resistance), and then mark with a superscript those ones which are different (by using Pearson).
RESPONSE: Table 3 (previous manuscript) has been redone, please seen the revised manuscript. The total AMR patterns of two populations were statistically compared using Pearson’s chi-squared test.
(8) L239-243 move to material and methods or rewrite, because is unclear.
RESPONSE: Line 239-243 (previous manuscript): the sentences have been revised to: “After the determination of homologous Salmonella isolates among paired individuals (serotype screening, and confirmed by PFGE and SR-FTIR), the two serotype-matched pairs: H121 & D121 and H86 & D86 were homologous isolates (Figure 3).” (line 245-247 of revised version)
(9) L 247-248: term relatedness proportion is inadequate. Authors mention “incidence” for the first time, and they do no explain how it is calculated nor the incidence at all, because it is a cross-sectional study and these type of studies do not measure incidence but proportions or prevalence.
RESPONSE: Line 247-248 (previous manuscript): the sentences have been revised, to: “The prevalence of homologous Salmonella pairs of the total Salmonella-present households was 5.13% (2/39).” (line 271-272 of revised version)
(10) L252-273: it should be in the manuscript not a figure note.
RESPONSE: Line 252-273 (previous manuscript): the note of the figure 3 have been moved to the result part, please see the revision. (line 248-268 of revised version)
3. Discussion part:
(1) L293-302: these are results
RESPONSE: Line 293-302 (previous manuscript): for a better discussion, the important results were summarized in the discussion part.
(2) L309-316: these sentences are introduction
RESPONSE: Line 309-316 (previous manuscript): the sentences on the “paired sample investigation” were designed for this special target. It is an infrequent cross-sectional study, which is worthy to be discussed.
(3) L316-320: no relevant (definition of a cross-sectional study). This study is a cross-sectional study with one study population (households), if not, well authors need to revise the whole manuscript and make clear why is not.
RESPONSE: Line 316-320 (previous manuscript): this work is not an investigation of the households. For a cross-sectional study on households, all the family members should be involved. However, the whole family member investigation would lead to several disadvantages 1): we cannot get the association between pet dog and the close contact caregiver. 2): the different numbers of households would exhibit the unexplainable information among many households 3): It would be difficult to explain the Salmonella passage among several experimental subjects in one household. To obtain the data on the Salmonella passage between pet dog and its most close contact caregiver, the paired sample investigation was the suitable experimental design. Although, there were still some limitations on the design, the true Salmonella passage rate would be lower than or equal to this estimated result. It still raised the doubts on that hypothesize: close contact between dogs and their caregivers was the main cause of Salmonella infection in Khon Kaen.
4. Minor points
(1) Title: there are some typos “Antimicrobial Resistance of Non-typhoidal Salmonella Isolates”
RESPONSE: Title: the title has been revised as “Risk of Antimicrobial Resistant Non-typhoidal Salmonella during Asymptomatic Infection Passage between Pet Dogs and their Human Caregivers in Khon Kaen, Thailand”
(2) Conclusion: do not include references
RESPONSE: The references in the conclusion part were removed.
(3) In addition:
Authors should be consistent with the terminology: pet dog or dog pet, caregivers or caretaker, AMR Salmonella or AMR patterns of Salmonella).
Use always the same number of decimals (one is enough for the level of precision used in this study)
Authors mention strain, sample and isolate indistinctly, I suggest the authors to be accurate
RESPONSE: The terminology has been revised: “caretakers” to “caregivers”. We had not mentioned “dog pet”. The AMR Salmonella is different to the AMR pattern of Salmonella.
RESPONSE: All the terminologies (samples, isolates, strains, and test) were checked and revised again.

Round 2
Reviewer 2 Report
The study has improved for better understanding. However, some comments still require clarification for clarity of readers.
L320: "pet dog source pathogenic AMR", the bacteria (or microorganism) are pathogenic. Please correct.
Methods: Sampling methodology is not well explained (which is important in prevalence studies). For example:
a) based on sample size calculation, 140 households were sampled (were 28 households per village?) Please complete.
b) later on the text and based on the sentence "140 households were randomly interviewed", it is infered that only 104 households were interviewed "For the 104 interviews". What did happen with the others 36 households?. Besides, L210-212 present results from 140 households.
Sentence L103-105 is incomplete.
Usually statistical analysed is carried out by tests (in this study the McNemar and Pearson tests) by using statistical software (SPSS in this study). L42-144 seems that the analyses was done by SPSS with P<0.05 significant. Please correct the paragraph.
L210-212: it is weird to read "Salmonella were present in households among individuals". Please update this sentence.
Table 2 is much clearer now that before. Thanks for updating. Could authors include also percentage (number of isolates with AMR/number of total isolates)?
Discussion:
L329: authors stated "this paired sample investigation seems like a variant of the cross-sectional study", what type of study is this article?
L330-331: readers could undestand that the outcome is the presence of Salmonella (or the AMR Salmonella), but what is it the exposure? Please explain.
I think this study looks like a cross-sectional study, where the study population are the caregivers (not all familiy members) and the dogs. In the methods section, sample size calculation is described for a cross-sectional study to estimate proportions within a dog population (n=73) or human population (n=138)
L341: change "owner" for "caregiver"
L342-343: could authors explain how the evaluations in other regions limit the results of this study?
L344-346: what is it "genuine prevalence"? Does the author mean true prevalence, which depends on the sensitivity and specificity of the test, study population among other factors? Please clarify.
Author Response
Dear Reviewer 2:
Thanks so much for your patient review, the manuscript has been much improved. We have addressed the issues according to these constructive suggestions. Below is a summary of the changes we performed and the responses to your comments and recommendations.
Sincerely,
Xin Wu
L320: "pet dog source pathogenic AMR", the bacteria (or microorganism) are pathogenic. Please correct.
RESPONSE: These words “pet dog source pathogenic AMR” has been revised to “pet dog source AMR Salmonella” (Line 320-321 of revised version)
Methods: Sampling methodology is not well explained (which is important in prevalence studies). For example:
a) based on sample size calculation, 140 households were sampled (were 28 households per village?) Please complete.
b) later on the text and based on the sentence "140 households were randomly interviewed", it is infered that only 104 households were interviewed "For the 104 interviews". What did happen with the others 36 households?. Besides, L210-212 present results from 140 households.
RESPONSE: a) The details of samples in each village were listed in the text, saying “One hundred and forty households were randomly interviewed from five villages (31 households from Bua Simma village, 24 from Mo Sum Soong village, 29 from Non Udom village, 44 from Noun Kung Soen village, and 12 from Nai Muang village) in Khon Kaen Province, Thailand,” (Line 94-96 of revised version)
- b) The number was wrong typing, 140 households were interviewed. The number has been revised.
Sentence L103-105 is incomplete.
RESPONSE: This sentence was removed, because we think this sentence was redundant.
Usually statistical analysed is carried out by tests (in this study the McNemar and Pearson tests) by using statistical software (SPSS in this study). L42-144 seems that the analyses was done by SPSS with P<0.05 significant. Please correct the paragraph.
RESPONSE: The paragraph has been revised saying that “To assay the epidemiological relatedness of Salmonella isolates from asymptomatic infection passage between the two populations, the presence of Salmonella-positive samples was statistically analyzed by McNemar's test; the serotypes and antimicrobial patterns of Salmonella isolates were statistically analyzed by Pearson’s chi-squared test. The statistical analysis were performed using statistical software SPSS version 17.0 (SPSS Inc, Chicago, Illinois, USA), P < 0.05 was considered significant.” (Line 141-146 of revised version)
L210-212: it is weird to read "Salmonella were present in households among individuals". Please update this sentence.
RESPONSE: The sentence has been revised to “The Salmonella species were identified from 43 individuals of 39 households” (Line 210 of revised version)
Table 2 is much clearer now that before. Thanks for updating. Could authors include also percentage (number of isolates with AMR/number of total isolates)?
RESPONSE: The table 2 has been revised. The explanation was added into the note of table 2, saying that “The antimicrobial sensitive isolates were not participated in the statistical analysis.” The ratio of AMR isolates to total isolates was mentioned saying that “The total AMR containing Salmonella isolates from pet dogs was 44.44% (8/18), and in human caregivers was 64.00 % (16/25).” (Line 205-206 of revised version)
Discussion:
L329: authors stated "this paired sample investigation seems like a variant of the cross-sectional study", what type of study is this article?
L330-331: readers could undestand that the outcome is the presence of Salmonella (or the AMR Salmonella), but what is it the exposure? Please explain.
I think this study looks like a cross-sectional study, where the study population are the caregivers (not all familiy members) and the dogs. In the methods section, sample size calculation is described for a cross-sectional study to estimate proportions within a dog population (n=73) or human population (n=138)
RESPONSE: This study is a cross-sectional study, whereas the paired sampling provided more information (such as the similarity and difference between two populations), comparing to the traditional cross-sectional study. Due to the special target (the quantitative presence of homologous Salmonella isolates of two populations) of this work, this experimental design was developed, and be mentioned again in discussion part. The exposures and outcomes of one survey exist at the same time, because the cross-sectional study cannot explain what statues of one patient (be in exposed or end of the disease) but the evaluation of the border of disease.
L341: change "owner" for "caregiver"
RESPONSE: The “owners” has been changed to “caregivers” (Line 20, 329 and 342 of revised version)
L342-343: could authors explain how the evaluations in other regions limit the results of this study?
RESPONSE: The living styles were quite different among regions. For instance, in some rural area people prefer to feed raw meat to dogs, whereas in cities pet owners feed commercial food to dogs. These living styles possibly could influence the results.
L344-346: what is it "genuine prevalence"? Does the author mean true prevalence, which depends on the sensitivity and specificity of the test, study population among other factors? Please clarify.
RESPONSE: the genuine prevalence mean the true value of the prevalence, this was revised according to the other reviewer who mentioned the bacteria culture method was limited. After deep consideration, the sentence needed to be revised, saying that “determining a higher confidence level of the Salmonella prevalence among the study participants, due to the limitation of bacterial cultures.” (Line 345-347 of revised version)

Round 3
Reviewer 2 Report
Thank you for the opportunity to review this paper. I think the revisions made have added greatly to the clarity and value of this paper. I am happy for it to be accepted for publication.
Author Response
Dear Reviewer 2:
Thank you for your kindly help
Best regards
Xin Wu